# OpenReview forum: "Unified Detoxifying and Debiasing in Language Generation via Inference-time Adaptive Optimization"
_ICLR.cc/2023/Conference — ICLR 2023 poster_

### Official Review · Reviewer_4XSk · 2022-10-28

**Confidence:** 4
**Correctness:** 3
**Technical Novelty And Significance:** 2
**Empirical Novelty And Significance:** 3
**Recommendation:** 8

**Clarity, Quality, Novelty And Reproducibility:**

Clarity, Quality:
Paper cites the relevant literature. However, as pointed out in the weaknesses section, some of the experiments are missing.

Novelty:
The proposed framework to unify detoxifying and debiasing is novel.

Reproducibility:
As pointed out in the weakness section, some critical details to reproduce the results are missing.



**Strength And Weaknesses:**

### Strengths:
- To the best of my knowledge, this is the first framework to unify debiasing and detoxifying tasks for open-ended generation.
- Authors have provided clear motivation and intuitions for most of the design choices. Authors have conducted ablation studies to show the effectiveness of their approach.

### Weaknesses:

- While the authors emphasize on unification, they use two different versions of attribute classifiers. In particular, for debiasing, they use embedding vectors’s principal component but for detoxifying, they use PPLM’s classifier. It’s not clear 1) why do we need two classifiers?  2) what gains do we get if we use PPLM’s classifier for both tasks.
- Authors suggest that their framework updates bias terms in only a few layers. However, for debiasing, they update bias terms across all layers. While they frame it as they do it because their framework is efficient, selecting a few bias terms to update is a non-trivial problem. If we select top-k layers for bias update, we might get different results. Authors don’t have any ablation study to show the effectiveness of selecting bias terms for a few layers.
- Authors mainly rely on perplexity as a proxy to measure quality of generations which is troublesome because 1) perplexity can’t be trusted as a generation quality metric. 2) authors use perplexity as a threshold. Authors present human evaluation results on a subset but do not show correlation with their automated metrics which makes it hard to understand how trustworthy the results are?
- Multiple critical details are missing from the paper which makes it very difficult for an individual to reproduce the results presented in the paper.
  - Authors construct their own dataset using publicly available BOLD dataset.  The exact dataset used for the experiments is not available in public which makes it difficult to reproduce the results presented in the paper.
  - A word can be splitted into multiple subwords, it’s not clear how it’s being handled at attribute classifier level.
  - Hyperparams used for generations are not defined. Recent works [1, 2] show that decoding algorithm hyperparameters alone play a big role on generated text fairness and quality so it’s important to explicitly define these and ideally use a range of hyperparameters instead of a fixed set of hyperparmas.

References:
-  [1] Akyürek, A. F., Kocyigit, M. Y., Paik, S., & Wijaya, D. (2022). Challenges in Measuring Bias via Open-Ended Language Generation. arXiv preprint arXiv:2205.11601.
-  [2]  Dhamala, J., Kumar, V., Gupta, R., Chang, K. W., & Galstyan, A. (2022). An Analysis of the Effects of Decoding Algorithms on Fairness in Open-Ended Language Generation. arXiv preprint arXiv:2210.03826


**Summary Of The Paper:**

This paper presents a unified framework for simultaneously detoxifying and debiasing language models. They consider protected groups as well as toxicity as an attribute of the generated text which can be controlled during the decoding time during an attribute classifier. Authors propose to use an embedding representation based attribute classifier for debiasing and a toxicity classifier for detoxifying tasks. In order to make debaising and detoxifying task parameters efficient, they propose to update only bias terms. Experiments show that the proposed method is computationally efficient and leads to minor degradation in the generation quality.

**Summary Of The Review:**

Debiasing and Detoxifying are very important problems for NLG community. While building solution to tackle bias and toxicity, researchers often treat these as two separate problems. I think this paper is a good contribution towards unification of debiasing and detoxifying solutions.

---

> ### Author Response · Authors · 2022-11-19
> **Thank you for the supportive review (1/3)**
>
> Thank you for your supportive review and suggestions. We have uploaded a revision of our paper.
>
> ***Question 1: Why do we need two classifiers for debiasing and detoxification when emphasizing "unification"? What gains do we get if we use PPLM’s classifier for both tasks?***
>
> The fundamental reason is that bias and toxicity are two different attributes. As stated in Section 3.1, bias reflects the different polarity of demographic groups (e.g., male and female), while toxicity is an inherent property. Therefore, in order to characterize these two different attributes, a separate classifier for each attribute would be the most appropriate. Besides, the emphasis of "unification" is to unify the optimization objective of debiasing and detoxification to simultaneously achieve these two goals and obtain an ethical NLG model. This is motivated by our observation that debiasing techniques alone achieve suboptimal detoxification performance compared with detoxifying techniques, while detoxifying techniques alone might even amplify social biases (see Figure 1-(b) and the third paragraph in the Introduction section).
>
> The classifier used in the PPLM work is constructed for toxicity classification. Using only this classifier in our framework leads to UDDIA-t. Table 5 shows the performance comparison between UDDIA-t (where only the PPLM's classifier is used) and UDDIA-u (where the two classifiers of both attributes are used). It is shown that UDDIA-u performs better in both detoxification and debiasing than UDDIA-t, suggesting the necessity of both classifiers.
>
> Another experimental challenge for building a PPLM-style classifier of the bias attribute is that there are no *off-the-shelf* annotated social bias datasets (mentioned in the **attribute classifier** part in the updated Section 3.3). Therefore, we resorted to seeking the embedding polarity as a reflection of the bias attribute and calculated the principal components by drawing inspiration from embedding debiasing methods. While our UDDIA framework supports any type of (bias/toxicity) attribute classifiers, we leave the advanced construction of them as our future work.
>
>
> ***Question 2: No ablation studies are shown to verify the effectiveness of selecting bias terms for a few layers. For debiasing, bias terms across all layers are updated.***
>
> For the scenario of detoxification, the proposed *redo* mechanism is dedicated to selecting optimal K when aiming to optimize the bias terms in top-K layers. *Redo* leverages a perplexity threshold $\mathbf{TH}$ to implicitly determine the number of the top layers to be tuned in an instance-wise manner. The six blue dots in Figure 3-(a) represent the effect of varied $\mathbf{TH}$s. Moreover, *the orange line in Figure 3-\(c\) also illustrates the ablation study of varied numbers (3,6,12, and 18) of the tuned top layers* without using *redo*. The ablation studies in Figure 3-\(c\) show that (i) the number of tuned top layers achieves a tradeoff between toxicity and fluency, and (ii) using *redo* leads to a better toxicity-fluency tradeoff than without using *redo*.
>
> For the scenario of debiasing, we added ablation studies on tuning the bias terms in the top-K layers of the 12-layer ${\rm GPT2}_{\rm base}$ with K=3,6,12, where K=12 is equivalent to tuning bias terms across all layers. The results are as follows:
>
> |     | Global Bias$\downarrow$ | Local Bias$\downarrow$ | Quality$\downarrow$ |
> | ------- | --------------- | -------------- | ----------- |
> | $K=12$    |  **2.13**  |   **8.79**  |  21.78  |
> | $K=6$     |  2.54  |  13.29 |  21.45  |
> | $K=3$     |  3.30  |  13.82 |  **21.31**       |
> | $K=12$-QM |  3.00  |  15.82 |  22.04  |
>
> *Quadratic mean of the metrics on the global bias, the local bias, and quality on gender. QM means updating only the parameters of the bias terms in the query and the second MLP layers, as proposed in (Ben Zaken et al., 2022). See the updated Table 13 for more detailed results.*
>
> It is shown that as the number of tuned bias terms increases, the debiasing performance is improved, but the generation quality deteriorates. The improvement of debiasing is significant, and quality loss is negligible when tuning the bias terms in all layers. Besides, as mentioned in the original Sec.3.4, our classifiers used in debiasing are highly efficient, and the generation quality is satisfactory. Therefore, we directly update the bias terms in all layers of the PLM during decoding. We added more ablation results and discussions in the revised Appendix E.1.2.

---

> > ### Author Response · Authors · 2022-11-19
> > **Thank you for the supportive review (2/3)**
> >
> >
> > ***Question 3: On measuring the quality of generations, perplexity is unreliable. The correlation of human evaluation results with the performance of automated metrics is not shown.***
> >
> > We agree that PPL is not a perfect metric of generation quality. However, we believe our results are trustworthy and could support the effectiveness of our model in generating fluency text for the following reasons:
> > 1. PPL is widely adopted to measure the fluency of generated text in open-ended NLG work (Welleck et al., 2020; Su et al., 2022). It's also a common practice to report PPL as a quality measure in most previous NLG debiasing/detoxification work (Bordia et al., 2019; Schick et al., 2021; Liu et al., 2021; Qian et al., 2022).
> > 2. We didn't use PPL as the *only* metric. For debiasing experiments, we also report the LM Score (Nadeem et al., 2021). For both debiasing and detoxification, we conduct human evaluations (with acceptable inter-annotator agreement). Different fluency metrics show consistent conclusions:  for debiasing, GPT2 > UDDIA-b >> A-INLP; for detoxifying, GPT2 ≈ UDDIA-t > DExperts. We believe these results, as well as the generated samples (Figures 2-(b),14,15,17), could verify the superiority of our model in generation fluency.
> > 3. We have also further calculated the correlation of human evaluation results with the corresponding perplexity scores for each sentence. The Pearson correlation coefficient is 0.41, which is similar to those reported by related work (Ke et al., 2022), indicating a moderate correlation (Mavuto M., 2012).
> > 4. We also simply tried another generation quality metric, MAUVE (Pillutla et al., 2021), on the text generated in debiasing experiments. We got consistent results on generation quality: UDDIA $\approx$ GPT-2 $>$ A-INLP (see the updated Appendix E.1.3 for details), which further verifies the reliability of our quality results. Note that we chose PPL as the threshold of *redo* just because we adopt it as the fluency measure. One could use any other quality metrics. We leave the further study of quality metrics for future work.
> >
> > We have included the above results and discussions in the revised Appendix E.1.3.
> >
> > References:
> > - Bordia et al. Identifying and Reducing Gender Bias in Word-Level Language Models. NAACL 2019.
> > - Schick et al. Self-Diagnosis and Self-Debiasing: A Proposal for Reducing Corpus-Based Bias in NLP. TACL 9(2021): 1408-1424.
> > - Liu et al., DExperts: Decoding-time controlled text generation with experts and anti-experts. ACL 2021.
> > - Qian et al. Controllable Natural Language Generation with Contrastive Prefixes. Findings of ACL 2022.
> > - Welleck, et al. Neural Text Generation With Unlikelihood Training. ICLR 2020.
> > - Su et al. A Contrastive Framework for Neural Text Generation. arXiv preprint arXiv:2202.06417 (2022).
> > - Ke et al., CTRLEval: An Unsupervised Reference-Free Metric for Evaluating Controlled Text Generation. ACL 2022.
> > - Mavuto M. A guide to appropriate use of correlation coefficient in medical research. Malawi medical journal 24.3 (2012): 69-71.
> > - Pillutla et al. MAUVE: Measuring the Gap Between Neural Text and Human Text using Divergence Frontiers. NeurIPS 2021.
> >
> > ***Question 4: The exact dataset constructed for the experiments is not available, which limits reproducibility.***
> >
> > We described our dataset construction in the original Appendix B.1 (the Data part) and B.2 (the Dataset construction part). We have added more details about how our dataset is constructed in the revised Appendix C.1 & C.2. We will also release all our constructed datasets used in our experiments for better reproducibility.
> >
> > ***Question 5: When the attribute classifier is processing, is it working on each subword or each word?***
> >
> > The attribute classifier processes each token at the sub-word level. Specifically, for debiasing, $p_{\omega}(a|x)$ is calculated over all $x$'s in the vocabulary list of the GPT2 models, which is at the subword level (See https://huggingface.co/gpt2/resolve/main/vocab.json). For detoxification, we directly use the toxicity classifier in PPLM. The classifier's input is the average of hidden states from the top layer of the GPT2 model. Since each hidden state is calculated by the forward propagation of each corresponding input token which is at the subword level, we can say that the toxicity classifier handles each token at the subword level as well. This processing (token-level classifiers) is consistent with most previous detoxification/debiasing work.

---

> > > ### Author Response · Authors · 2022-11-19
> > > **Thank you for the supportive review (3/3)**
> > >
> > > ***Question 6: Explicitly define the hyperparameters used for generations.***
> > >
> > > We explicitly defined the decoding hyperparameters and added more details in the updated Appendix C.1 (Settings). We also cited the two papers you suggested and included discussions about the influence of decoding hyperparameters in the updated Appendix E.1.4. In summary, we believe our experimental results could well support the effectiveness of our model for the following reasons:
> > >
> > > 1. All baselines in debiasing experiments share the same decoding hyperparameters. Therefore, our comparisons are fair, and there is neither the risk of the flipping bias tendency in (Akyürek et al., 2022) nor the misleading conclusion due to different decoding setups discussed by (Dhamala et al., 2022).
> > > 2. We report multiple biases and quality metrics (and their quadratic mean) to avoid experimentally biased evaluations and manifest consistent improvement by our model. Besides, the two metrics from StereoSet (Nadeem et al., 2021), namely, ICAT and LM Score, are based on PLM output distributions, irrelevant to decoding methods. Therefore, the discrepancy of different sentence-based metrics (e.g., Regard and sentiment) is not a big issue in our experiments.
> > > 3. For each prompt, the reported value on each metric is the average of all sampled continuations (100 for each prompt). Therefore, there is *no* risk that 'particular samples considered may affect' as suggested in (Akyürek et al., 2022).
> > > 4. We follow the hyperparameter settings of previous works (Sheng et al., 2020). Most of the settings (e.g., top-p, p=0.9, top-k, k=40) are also common practices in various open-ended language generation applications.
> > > 5. We also simply tried different lengths of generated continuations on gender bias and observed consistent debiasing effectiveness compared to the original GPT-2. See the updated Appendix E.1.4. for more results. We leave further analysis of the influence of decoding settings for future work.
> > >
> > > ***Question 7: Authors say “our metric is quantifying a different notion of fairness issues compared to the existing metrics”. However, they don’t define the “fairness notion” that they are trying to measure.***
> > >
> > > We find that the sentence “our metric is quantifying a different notion of fairness issues compared to the existing metrics” does not exist in our paper. Is this question meant for others' submission?

---

> > ### Comment · Reviewer_4XSk · 2022-12-03
> > **Thanks for the additonal experiments**
> >
> > Thank you for answering my questions and providing the updated draft. I believe it's a good contribution towards unification of debiasing and detoxifying solutions. I have updated my score to reflect that.

---

> > > ### Author Response · Authors · 2022-12-04
> > > **Thank you again**
> > >
> > > Thank you for updating the score! Your suggestions have really helped us improve our work. We will further revise our paper with the added experiments and discussions.

---

### Official Review · Reviewer_Syvk · 2022-11-02

**Confidence:** 3
**Correctness:** 3
**Technical Novelty And Significance:** 3
**Empirical Novelty And Significance:** 3
**Recommendation:** 8

**Clarity, Quality, Novelty And Reproducibility:**

There are clarity issues in many places as described in the previous section

The method appears novel as far as I am aware of this work. And seems like the results can be reproduced given hyperparameter details.

**Details Of Ethics Concerns:**

This work presents a method of debiasing and detoxification. They also provide an ethics statement listing limitations and future work, which should be reviewed and verified.

**Strength And Weaknesses:**

Strengths:
1. Substantial empirical improvements over baselines, especially in the debiasing setup which are confirmed by human evaluation.
2. Most individual losses proposed are simple and straightforward to implement.

Weaknesses:
1. The definition of bias is not clear. In 3.1, bias is defined as the difference between f(.) value of two outputs xi and xj generated by different prompts ci and cj. It is not clear if this property is being measured per example or distributionally (the latter makes more sense but it is not clarified in the writing). Additionally, why should it to be two different prompts with biased outputs that should be considered. Shouldn't the goal be given a single prompt, the output should equally likely predict outputs related to either class (gender or race)?
2. In 3.5, a threshold TH is defined to control for perplexity. This threshold in practice is seems difficult to define properly in realistic settings since given the context length and the output sequence length, the perplexity can vary a lot.
3. The writing is confusing in many places. Multiple definitions of bias is being used without clarification (social bias vs the bias term in neural networks). confusing notations with $x$ used to refer to both whole sequences and just tokens.

Questions:
1. What is the intuition behind using Hellinger distance instead of other distances?
2. For evaluating global bias, how are ci and cj pairs selected? Are there any restrictions placed on whether the prompts talk about the same topic/content?

**Summary Of The Paper:**

This paper presents a decoding algorithm that reduces (racial, gender) bias and toxicity in the generated outputs of language models. The proposed algorithm consists of token-level and context-dependent components that modify the bias terms of selected or all the layers of a language model at decoding time to suppress or encourage desired behaviors measured by attribute classifiers. Applied to reduce bias between different pairs of prompts and toxicity in output text, the author report improvements over standard evaluation measures for control and quality of the output text.

**Summary Of The Review:**

The presented algorithm for debiasing and detoxifying language model outputs is simple to implement and has sound motivations. There are some clarity issues which make the paper hard to follow, especially in section 3. This should be addressable in future versions of this paper, and I am willing to revise my score if my questions are answered and clarifications provided.

---

> ### Author Response · Authors · 2022-11-19
> **Thank you for the supportive review (1/3)**
>
> Thank you for your supportive review and kind suggestions. We have uploaded a revision of our paper.
>
> ***Question 1: The definition of bias is not clear. Is the defined bias metric measured per example or distributionally? Why using two different prompts to measure bias instead a single prompt?***
>
> 1.1. *Whether the social bias is being measured per example or distributionally.*
>
> The social bias reflects the *distribution-level* difference of PLMs between different demographic groups and *is being measured distributionally*. We have improved the definition of social bias from the distributional perspective in the updated Section 3.1.
>
> Specifically, consider a sensitive attribute $a$ (e.g., gender), which is assumed to be binary, e.g., male ($a=0$) and female ($a=1$). The gender bias of a PLM is reflected as the difference of $p_\theta(x|a=0)$ and $p_\theta(x|a=1)$. Following (Dwork et al., 2012), we use total variation, $D_{TV}(p_\theta(x|a=0),p_\theta(x|a=1))$, to characterize such a difference.
>
> In practice, the text $x$ is often generated from a given prompt $c$. Therefore, we incorporate the prompt and have $p_\theta(x|a)=\int p_\theta(x,c|a) dc=E_{p(c|a)}[p_\theta(x|a,c)]$. Since $D_{TV}(p_0,p_1)=\frac{1}{2}\sum_x\lvert p_0(x) - p_1(x) \rvert$, we get $D_{TV} = \frac{1}{2}\sum_x\lvert E_{p(c|a=0)}[p_\theta(x|a=0,c)] - E_{p(c|a=1)}[p_\theta(x|a=1,c)] \rvert$. Concretely, we have $D_{TV}\approx \sum_x \frac{1}{2M}\lvert \sum_m p_\theta(x|c^0_m) - p_\theta(x|c^1_m)\rvert$, with $c^0_m\sim p(c|a=0)$, $c^1_m\sim p(c|a=1)$ and $M$ as the number of prompts. Given this, we approximate $p(c|a)$ by constructing attribute-conditioned prompts with the mentioning of specified demographic groups. For example, we have $c^0_m$=''The *man* was known for'' for male, and $c^1_m$=''The *woman* was known for'' for female (see more examples in Appendix A). Based on these formulations of the total variation, the two types of bias measures in the original paper can also be induced from the distributional perspective:
> - For *local bias*, we have $D_{TV}\approx \sum_x \frac{1}{2M}\lvert \sum_m p_\theta(x|c^0_m) - p_\theta(x|c^1_m)\rvert$;
> - For *global bias*, we have $D_{TV}\approx \sum_{(x^0, x^1)}\frac{1}{2M}\sum_{m}\lvert \sum_m p(h|x^0_m) - p(h|x^1_m)\rvert$ , with $x^0_m \sim p_{\boldsymbol\theta}(x|c^0_m)$ and $x^1_m \sim p_{\boldsymbol\theta}(x|c^1_m)$.
>
> See the detailed descriptions and derivations in the updated Section 3.1 and Appendix A. We also appreciate this question: To the best of our knowledge, we are the first to draw the formal connection between the local bias, the global bias, and total variation to provide interpretations from a distributional perspective.
>
> 1.2. *Shouldn't the goal be given a single prompt, the output should equally likely predict outputs related to either class?*
>
> When conducting social bias evaluation, a randomly selected prompt $c$ cannot always steer attribute (e.g., gender) related continuations. For example, taking the prompt $c$=''I like to eat'' as input, PLMs often generate unbiased continuation. Since most of such prompts are meaningless for bias evaluation and it's infeasible to traverse all valid prompts, we adopt *counterfactual evaluation* (Huang et al., 2020) to construct prompt pairs that explicitly mention different demographic groups to encourage attribute-relevant generations.
>
> This evaluation protocol is also mathematically equivalent to the single-prompt evaluation that you proposed. Consider that conditioned on a single prompt $c$, $p(a|x,c)$ characterizes the probability that the attribute $a$ is expressed by the generation $x$. Using Bayes' rule, we have $p(a|x,c)=\frac{p(x|c,a)p(a,c)}{p(x,c)}\propto p(x|c,a)$, as we can omit $p(a,c)$ and $p(x,c)$ since given $a$ and $c$, $p(a,c)$ is a constant and $p(x,c)$ is irrelevant to bias. At the same time, our evaluation protocols focus on $p(x|c,a)$. Therefore, it can be seen that the two evaluation protocols are theoretically equivalent.
>
> References:
> - Dwork et al. Fairness through awareness, ITCS 2012.
> - Huang et al., Reducing sentiment bias in language models via counterfactual evaluation. Findings of EMNLP 2020.

---

> > ### Author Response · Authors · 2022-11-19
> > **Thank you for the supportive review (2/3)**
> >
> > ***Question 2: The threshold TH is difficult to define properly in realistic settings, since the perplexity can vary a lot.***
> >
> > The threshold **TH** can be interpreted as a hyperparameter in our adaptive *redo* mechanism. Such a hyperparameter allows a flexible trade-off between toxicity and generation fluency, and previous detoxifying and debiasing methods also widely make use of a similar hyperparameter (e.g., $\alpha$ in DExperts and A-INLP) for such a trade-off. The tunable hyperparameters in fact bring more freedom in terms of controllability. Illustrated in Figure 3-(a), the DAPT method, without a similar hyperparameter for trade-off, achieves suboptimal fluency and toxicity performance. Figure 3-(a) also shows that **TH** in our *redo* generally achieves a better trade-off than the $\alpha$ in DExperts, especially when better fluency is required.
> >
> > In realistic settings, one can tune the **TH** hyperparameter to adapt to the sentences to be processed. A straightforward reference point for **TH** can be obtained by estimating the perplexity of the generations produced by the original GPT-2 model (given the context length and the desired output sequence length). For example, in our experiments, we follow the settings used in previous works to use contexts from RealToxicityPrompts and set generation length = 20 for detoxifying experiments. The original GPT-2 model produces generations with overall PPL = 25.45. Therefore, in order to produce detoxified generations without sacrificing too much on fluency, one could set the perplexity threshold **TH** in *redo* to be around 25.45, or slightly larger since not every generation process triggers *redo*. In fact, many generations are of less perplexity score than **TH** in their first runs, so the resulting perplexity score averaged over all generations is usually less than **TH**. As shown in Table 3, setting **TH**=40 results in PPL = 26.92, and setting **TH**=30 results in PPL = 22.64.
> >
> > We have also added the above discussion on tuning **TH**s in the revised Appendix E.2 as practical suggestions for the readers who try to apply UDDIA-t/UDDIA-u and the *redo* mechanism in their applications.
> >
> > ***Question 3: Confusion in writing. For example, the usage of the word "bias" and the notation "x".***
> >
> > We improved the clarity of the terms in our revised paper. For all the "bias"s that refer to the bias terms in a model, we use the full term "bias term". We also added a footnote in Page 4 as a reminder to readers that "not to be confused with the social bias". For the notation "x", we use the bold symbol $\mathbf{x}$ to represent a sentence and the normal type $x$ as a token in our updated paper (by the way, we also wished to write in this format in the rebuttal, but there seems to be an issue in displaying complex equations correctly in the OpenReview comment system...). We have also introduced the notation of time step $t$ and used $x_t$ to denote the $t$-th token in a sentence whenever necessary.
> >
> > ***Question 4: The intuition behind using Hellinger distance instead of other distances.***
> >
> > We select Hellinger distance because it is bounded within [0,1]. This property makes it more suitable as a metric that measures local bias, and it can be easily scaled to [0,100] to be combined with the other local bias metric ICAT. Besides, we also follow the previous work, e.g., A-INLP (Liang et al., 2021), which adopted Hellinger distance as the local bias metric.

---

> > > ### Author Response · Authors · 2022-11-19
> > > **Thank you for the supportive review (3/3)**
> > >
> > > ***Question 5: How are ci and cj prompt pairs selected and constructed to evaluate global bias?***
> > >
> > > As we described in the original Appendix B.1, $c_i$ and $c_j$ are carefully-designed and created to share most (attribute-irrelevant) content except the demographic mentions. For example, for gender, $c_i=$''The *man* was known for'' and $c_j=$''The *woman* was known for''. Therefore, the two prompts $c_i$ and $c_j$ contain the same content but the different mentioning of demographic groups (see Table 8 for more examples).
> > >
> > > More concretely, we consider two kinds of prompts, namely, simple prompts and diverse prompts:
> > >
> > > 1. *Simple prompts*. Prompts constructed from manually designed templates like '*The $A$ had a job as*' and '*The $A$ earned money by*' where $A$ is the demographic group placeholder. For each template, we replace $A$ with different values (groups) in the same attribute. For example, for gender, we replace $A$ with 'man' and 'woman'; for race, 'black person' and 'white person'. The template is neutral and could motivate PLMs to generate descriptive continuations of the given demographic group. We directly use the templates provided in (Sheng et al., 2019; 2020).
> > > 2. *Diverse prompts*. As discussed in (Liang et al., 2021), LMs must handle many possible diverse contexts and show no social bias in generated continuations. The simple templates used in simple prompts may fail to cover the variety in context and lose context associations. To evaluate the debiasing ability of different models in rich real-world contexts, we also construct diverse prompts. The construction process is similar to that of simple prompts, but we use more diverse sentences as templates, e.g., 'The $A$ is an American former professional'. In detail, we use the prompts in BOLD (Dhamala et al., 2021). However, the original group indicators in BOLD are all names, which would be too implicit. Therefore, we replace names with our more explicit demographic mentions.
> > >
> > > We have added more details of prompt construction in the updated Appendix C.1 and improved the descriptions of prompts in Sec. 4.1.1. We will also release our prompts.

---

> > > > ### Comment · Reviewer_Syvk · 2022-12-11
> > > > **Thank you for the response. Updated my score**
> > > >
> > > > I thank the authors for their response and all the edits they made to the paper. It has improved my understanding of the methodology. I think some of the desiderata that the debiasing is distributional and why it's essential should be clarified early on in the introduction. I am updating my score to 8.

---

> > > > > ### Author Response · Authors · 2022-12-12
> > > > > **Thank you again**
> > > > >
> > > > > Thank you for updating the score! Your advice really helped us improve our work. We will further polish our paper by clarifying the distribution-level definition in the introduction.

---

### Official Review · Reviewer_HmKe · 2022-11-02

**Confidence:** 3
**Correctness:** 3
**Technical Novelty And Significance:** 3
**Empirical Novelty And Significance:** 3
**Recommendation:** 5

**Clarity, Quality, Novelty And Reproducibility:**

Clarity - Clarity of the paper is good until Section 3.2
Quality - The paper is of reasonably high quality barring presentation.
Novelty - The approach is not extremely novel, but is novel enough and borrows interesting ideas from parameter-efficient training
Reproducibility - The paper alone certainly lacks the details needed to reproduce this work.

**Strength And Weaknesses:**

Strengths:
1. The paper seeks to solve an important problem, and is well written and motivated until about Section 3.3.
2. The approach, although quite expensive at inference time, avoids re-training the network.
3. The token-level attribute classifier for debiasing based on similarities with the first principal components of attribute-specific terms is clever.
4. Results on detoxification against DExperts which is a fairly strong baseline are impressive.

Weaknesses:
The main weakness of this paper is its presentation. The paper's intro, related work, and motivation around the use of mutual information minimization as a tool to do both debiasing as well as mitigate toxic responses are great, but the paper after that seems to lose focus and branches off into two fairly different approaches for debiasing and detoxification. These are my concerns about the presentation and structure

1. Reading appendix section D.2 was critical to my understanding of how the "redo" approach works - I would recommend moving it to the main body of the paper.

2. Having a pseudo-code block detailing the overall approach after all of the sections (after 3.5) will help the presentation. Right now it's not clear how each component comes together. Specifically what components do detoxification and debiasing share and what is different? Is one a token-level intervention but the other adopts an approach that re-generates things?

3. It is unclear how your initial motivation around minimizing mutual information I(x;a) ties into the final formulation for both section 3.4 and 3.5. There seems to be a disconnect in the presentation between sections 3.2, 3,3, and 3.4.

4. The paper seems to be missing clear definitions of $p_{\theta}$ and $p_{\omega}$ and their parameterizations.

Clarification:
In Section 3.3, shouldn't we be comparing P(x|c) and P(x|a,c) rather than P (x|c) and P(x|a)? Why is context conditioning missing when conditioning on an attribute? You mention an approach for context-aware rectification but you do not seem to use it in the subsequent formulation.

**Summary Of The Paper:**

The paper presents an approach to detoxifying and debiasing the outputs for language models at inference time with per-sample-based parameter-efficient fine-tuning. The paper takes two slightly different approaches to debiasing and detoxification but motivates them as being part of a unified Mutual Information Minimization framework between a generated token "x" and an attribute "a" (ex: male/female for debiasing or toxic/non-toxic for detoxification).

Debiasing takes an adaptive token-level intervention approach wherein parameter-efficient fine-tuning is done to the model when there is a large margin between p(x|c) and p(x|a,c). Optimization is done to minimize this gap based on a MI minimization objective. Detoxification, in contrast, takes an approach similar to PPLM (Dathathriet al., 2020), but adapts only a small fraction of the top layers of the network.

**Summary Of The Review:**

The paper has interesting ideas around the use of parameter-efficient tuning at inference time only, but would significantly benefit from re-organization and better presentation of the main ideas.

---

> ### Author Response · Authors · 2022-11-19
> **Thank you for the valuable review (1/2)**
>
> Thank you for your valuable review and suggestions. We have uploaded a revision of our paper.
>
> ***Question 1: Move Appendix D.2 to the main paper for better presentation of "redo" mechanism.***
>
> We added the description of *redo* in the original Appendix D.2 into the revised main paper for better presentation (see the updated Section 3.3).
>
> ***Question 2: Have a pseudo-code block after Sec. 3.5 to detail the overall approach and specify the shared and separate components for debiasing and detoxification.***
>
> We added a pseudo-code block in the updated Section 3.3 to abstract the shared framework for both debiasing and detoxification scenarios. Yes, the debiasing approach leverages the token-level adaptive intervention, and the detoxifying approach uses *redo*. For unified debias and detoxification, we use both *redo* and the token-level intervention strategy. We refactored the presentation of the design of the adaptive mechanisms into the updated Section 3.3 and also provided a detailed algorithm pseudo-code block in Appendix B (due to page limit).
>
> ***Question 3: How the initial motivation of minimizing mutual information (Sec. 3.2) ties into the final formulation for Secs. 3.4 and 3.5.***
>
> The mutual information minimization (Eq. (3)) in Section 3.2 is our objective for the debiasing, the detoxifying, and the unified experiments. For debiasing, we directly minimize Eq.(3) using the constructed attribute classifier. For detoxifying, as we are using the PPLM classifier, we leverage the loss terms used in the PPLM work. Nevertheless, we show that the PPLM loss agrees with our aim of optimizing Eq. (3) under certain assumptions (see the added Appendix D.5).
>
> We have improved the presentations in the updated Section 3.3. For clarity, we also added the pseudo-code for the abstracted UDDIA framework as Algorithm 1, and a more detailed Algorithm 2 in Appendix B.
>
> ***Question 4: Clear definitions of*** $p_\theta$ ***and*** $p_\omega$ ***and their parameterizations.***
>
> We added their definitions in the revised paper (see the colored words in the updated Section 3.2).
>
> ***Question 5： In Section 3.3, shouldn't we be comparing P(x|c) and P(x|a,c) rather than P(x|c) and P(x|a)?***
>
> We believe that you are referring to Hellinger distance $H$ in the original Section 3.4 (debiasing design) for our token-level adaptive intervention strategy. In this part, the "x" in "P (x|c) and P(x|a)" should represent the current ($t$-th) token $x_t$, instead of a sentence; We apologize for the confusing notations. We have improved the presentation in the revised paper for clarity.
>
> Below we detail the comparisons of the token-level intervention strategy and the context-aware rectification in the debiasing design:
>
> 1. *Intervention strategy (originally in Sec.3.4, now in the updated Sec.3.3)*. To determine at which time step $t$ to intervene, we need to measure the attribute polarity of the current LM output distribution $p_\boldsymbol{\theta}(x_t | x_{<t}, c)$ (e.g., whether predicted high-probability tokens are all gender-correlated). For this purpose, we calculate the Hellinger distance $H(p_\boldsymbol{\theta}(x_t | x_{<t}, c), p(x_t|a_k=v))$ where $p(x_t|a_k=v)$ is a prior distribution indicating all tokens correlated to the given attribute value $a_k=v$. For example, $p(x_t|a=\text{female})$ means all female-correlated tokens. A small $H$ indicates the current LM prediction is highly overlapped with the gender-correlated tokens (e.g., she, girl, nurse), which suggests that we need to intervene and rectify the distribution. While $p(x_t|a_k=v)$ is unknown, we approximate it with our attribute classifier using Bayes' rule: $p(x_t|a_k=v) \propto p_{\boldsymbol{\omega}}(a_k = v|x_t)*\hat{p}(x_t)$. Since $p(x_t|a_k=v)$ represents our *prior knowledge of socially-biased tokens in the vocabulary*, we directly used it and didn't consider the context to determine *when to intervene*.
> 2. *Context-aware Rectification (in Sec.3.2)*. Once we decide to intervene at the $t$-th time step according to $H$ above, we optimize the parameters using Eq.(3). The $p_{\omega}(a|x_t, x_{<t}, c)$ term in Eq.(3), where context is involved, naturally helps us tell whether the attribute polarity comes from the current predicted token $x_t$ itself or the context as discussed in Sec.3.2. The context is required here to calculate the context-aware mutual information loss (rather than the intervention strategy).
>
> In addition, we also conducted ablation studies on the intervention strategy by integrating the context into Hellinger distance calculation as $H(p_\boldsymbol{\theta}(x_t | x_{<t}, c), p(x_t | a,x_{<t}, c))$. The results on simple gender prompts are shown in the following table: (see the response (2/2))

---

> > ### Author Response · Authors · 2022-11-19
> > **Thank you for the valuable review (2/2)**
> >
> > (Following response (1/2)) The results on simple gender prompts are shown in the following table:
> >
> > |   Model      | R.$\downarrow$ | S.$\downarrow$ | T.$\downarrow$| Q.$\downarrow$| H.$\downarrow$ | I.$\uparrow$| Q.$\downarrow$| P.$\downarrow$ | L.$\uparrow$| Q.$\downarrow$
> > | :------- | ---------------: | --------------: | -----------: | ---------: |  ---------: | ---------: | ---------: | ---------: | ---------: | ---------: |
> > | GPT-2 | 5.57 | 2.89 |     1.62  |  3.74  |  16.14 | 81.79 | 17.21 | 10.77 | 68.85 | 23.31 |
> > | UDDIA-b | 3.54  | **0.79** | **0.63** | **2.13** | **8.93** | **91.36** | **8.79**  | **12.27** |**71.75**|**21.78**|
> > | UDDIA-b, w/ context in Hellinger | **0.81** | 4.00 | 2.73 | 2.83 | 13.52 | 89.34 |  12.17      | 12.33 |  65.69 |  25.78 |
> >
> > It is shown that incorporating context into the calculation of Hellinger distance $H$ even hurts the performance (with increased computation cost). The reason lies in that $p(x_t | a,x_{<t}, c)=\frac{p(a|x_t,x_{<t}, c)p_{\boldsymbol{\theta}}(x_t|x_{<t}, c)}{p(a|x_{<t}, c)}$. By comparing $p_{\boldsymbol{\theta}}(x_t | x_{<t}, c)$ and $p(x_t | a,x_{<t}, c)$, we are actually comparing $p(a|x_t,x_{<t}, c)$ and $p(a|x_{<t}, c)$. Because the representations of $x_t$, $x_{<t}$ and $c$ are linearly combined in our classifier (see updated Sec. 3.3), *the only different contribution to $a$ originates from $x_t$*, which has been reflected by the token-level intervention strategy. Moreover, the attribute polarity $p_{\boldsymbol{\omega}}(a|x_t,x_{<t}, c)$ would be obstructed by context compared to the original $p_{\boldsymbol{\omega}}(a|x_t)$. As a result, neutral or longer context would dilute the attribute polarity and lead to omitted debiasing steps (increased bias); attribute-related context would cause spurious polarity (decreased quality). For example, consider the sentence ''[The woman was well-known for her] job'': The context ''The woman was well-known for her'' has high gender polarity, but the generated continuation ''job'' is gender-neutral. In terms of $p_{\boldsymbol{\omega}}(a={\rm female}|c, x_t={\rm job})$, the model may unnecessarily intervene and mistakenly change the word ''job''. Therefore, we use a token-level intervention strategy without including the context $c$ in the Hellinger distance calculation.

---

### Author Response · Authors · 2022-11-30
**Looking forward to further feedbacks**

Dear Reviewers,

Thank you again for your valuable comments and suggestions, which are really helpful for us. We have posted responses to the detailed concerns.

We totally understand that this is a quite busy period, since the reviewers may be responding to the rebuttal of other assigned papers.

We deeply appreciate it if you could take some time to return further feedback on whether our responses solve your concerns. If there are any other comments, we will try our best to address them.

Best,

The authors

---

### Decision · Program_Chairs · 2023-01-20

**Decision:**

Accept: poster

**Justification For Why Not Higher Score:**

I think the topic is interesting yet slightly marginal, this is why I am recommending Acceptance as a poster.

**Justification For Why Not Lower Score:**

Very good author response, reviewers comments were mostly positive.

**Metareview: Summary, Strengths And Weaknesses:**

The paper presents an approach to detoxifying and debiasing the outputs for language models at inference time with per-sample-based parameter-efficient fine-tuning. The paper takes two slightly different approaches to debiasing and detoxification but motivates them as being part of a unified Mutual Information Minimization framework between a generated token "x" and an attribute "a" (ex: male/female for debiasing or toxic/non-toxic for detoxification). Reviewers had mostly clarification and presentation issues which the authors have addressed in their response. It is an interesting paper and should be presented at the conference.

This paper was flagged for ethical review, and the recommendation from the ethics committee was that the authors should explicitly recongize their assumptions with respect to the terms they employ (e.g., bias, race and toxicity) and be explicit about the limitation of the research. We expect the authors to respect these recommendations.

**Note From Pc:**

if the above contains the word "oral" or "spotlight" please see: "oral" presentation means -> notable-top-5% and "spotlight" means -> notable-top-25%. As stated in our emails, we are disassociating presentation type from AC recommendations